# The Design, Characterization and Antibacterial Activity of Heat and Silver Crosslinked Poly(Vinyl Alcohol) Hydrogel Forming Dressings Containing Silver Nanoparticles

**DOI:** 10.3390/nano11010096

**Published:** 2021-01-04

**Authors:** John Jackson, Helen Burt, Dirk Lange, In Whang, Robin Evans, David Plackett

**Affiliations:** 1UBC Faculty of Pharmaceutical Sciences, 2045 Wesbrook Mall, UBC, Vancouver, BC V6T1Z3, Canada; helen.burt@ubc.ca (H.B.); inwhang2010@gmail.com (I.W.); davidplackett@gmail.com (D.P.); 2Stone Centre, UBC Department of Urologic Sciences, Vancouver General Hospital, Vancouver, BC V6T1Z3, Canada; dirk.lange@stonecentrevgh.ca; 3Plastic and Reconstructive Surgery Clinic, Ventura county medical clinic, Ventura, CA 93003, USA; robinjamesevans@gmail.com

**Keywords:** wound dressing, silver nanoparticles, PVA hydrogel, anti-infective

## Abstract

The prompt treatment of burn wounds is essential but can be challenging in remote parts of Africa, where burns from open fires are a constant hazard for children and suitable medical care may be far away. Consequently, there is an unmet need for an economical burn wound dressing with a sustained antimicrobial activity that might be manufactured locally at low cost. This study describes and characterizes the novel preparation of a silver nitrate-loaded/poly(vinyl alcohol) (PVA) film. Using controlled heating cycles, films may be crosslinked with in situ silver nanoparticle production using only a low heat oven and little technical expertise. Our research demonstrated that heat-curing of PVA/silver nitrate films converted the silver to nanoparticles. These films swelled in water to form a robust, wound-compatible hydrogel which exhibited controlled release of the antibacterial silver nanoparticles. An optimal formulation was obtained using 5% (*w*/*w*) silver nitrate in PVA membrane films that had been heated at 140 °C for 90 min. Physical and chemical characterization of such films was complemented by in vitro studies that confirmed the effective antibacterial activity of the released silver nanoparticles against both gram positive and negative bacteria. Overall, these findings provide economical and simple methods to manufacture stable, hydrogel forming wound dressings that release antibiotic silver over prolonged periods suitable for emergency use in remote locations.

## 1. Introduction

Traumatic burn wounds resulting from accidents associated with open fires used for heating and cooking are common in the rural areas of African countries. Children in particular suffer many such injuries and, in remote locations far from medical care, burn wounds can often lead to more complex care situations [1,2]. Commercial wound dressings are normally much too expensive to be used in most African settings. We therefore initiated a research project aimed at developing a package or kit that could be used for immediate treatment of burn wounds in remote African communities. Such kits might contain a dressing, a pictorial instruction manual for application of the dressing, a bottle of rehydration salts, and cell phone numbers for clinics or nurses. These kits might provide an important window of opportunity for the injured person to access proper hospital care before serious infection or abnormal healing might occur.

The wound dressing, as the key component of the proposed burn survival kit, should provide an anti-infective comfortable wound covering for one week or more and be easily removable without trauma to the patient.

The properties of silver as a disinfectant have been known for centuries [3,4,5] but the wider use of silver as an antibiotic has only come about in recent decades, largely because of increasing concerns about the rise of infections caused by pathogenic bacteria that have, over time, become more resistant to commonly used antibiotics. In Western countries topical silver sulfadiazine creams, along with silver-based wound dressings such as Acticoat™ (Smith and Nephew) and Aquacel Ag^®^ (Convatec), are often used for treatment of long-term skin wounds. As confirmed by Khundkar et al. [6], Acticoat™ is now widely used, although the clinical data for its use are not strong. However, it is generally accepted that this product provides an improved antibiotic effect when compared to other silver-containing dressings. Two versions of this product are available as Acticoat™ 7 and Acticoat™ Moisture Control, both of which contain Silcryst™ silver nanocrystals [5]. The efficacy of Aquacel Ag^®^, based on sodium carboxymethyl cellulose fibers integrated with ionic silver, which forms a gel on wound contact, has been demonstrated through both in vitro and in vivo studies [7]. In terms of their practical use, these expensive wound dressings are generally changed every three days, but Acticoat™ may be left on a wound area for as long as a week. This paper describes research on the development of an inexpensive silver-based hydrogel dressing material that might provide a moist comfortable covering and provide some level of antibacterial protection from the silver component.

Poly(vinyl alcohol) (PVA) is a relatively inexpensive, biocompatible polymer that has been extensively studied for a wide range of medical applications and is used in commercial medical devices [8,9]. Due to solubility issues, for extended use, a cross-linked PVA gel would be preferable. Although there are reports in the literature on the use of borates and heat to cross-link PVA in a non-toxic manner [10,11], these methods do not provide long-term resistance to aqueous solubilization. A cross-linked water insoluble, yet hydrated, PVA film was considered a suitable formulation since its use could allow controlled release of an incorporated antibiotic over time rather than the immediate release of all available antibiotic that would likely occur from non-crosslinked PVA. We therefore pursued the idea of using silver both as an antibiotic and also as a hydrothermal cross-linking agent for PVA. Although their research was not aimed at a medical application, Luo et al. [11] demonstrated that silver-cross-linked PVA nanocables could be produced by a simple solution approach involving autoclaving at 160 °C for up to 72 h. We explored a similar approach that used lower temperatures and shorter times to facilitate both silver cross-linking of PVA films and in situ synthesis of silver nanoparticles. The antibiotic properties of silver nanoparticles have been widely reported [12,13,14] with recent evidence indicating that their antimicrobial efficiency is probably related to the slow release of silver ions rather than an intrinsic property of the nanoparticles per se [15]. It therefore seemed feasible that using cross-linked PVA films containing silver nanoparticles created in situ might meet the material needs of the burn wound dressing project. 

In recent years, a number of authors have reported methods for the preparation of PVA-silver nanocomposite films and have characterized the antibacterial and other properties of these films. Cencetti et al. [16] generated such films using a mixture of PVA with gellan, borax and silver nitrate in a hydroalcoholic formulation. The films showed sustained silver release and slow dehydration rates as well as antibacterial activity against *Staphylococcus aureus* and *Pseudomonas aeruginosa.* Another approach was adopted by Herron et al. [17] in which PVA-polyelectrolyte multilayer (PEM) films were formed by a layer-by-layer (LBL) technique to provide silver nanoparticle containing films with good antibacterial effects at low silver loadings in wound models. Oliveira et al. [18] specifically designed burn wound dressings by forming PVA-silver nanocomposite films in which silver nanoparticles were created by exposure to Co^60^ γ-radiation. The antibacterial activity of the films was confirmed by a disc diffusion method. Mahmoud [19] used sodium borohydride reduction of silver nitrate to create silver colloids that were then combined with PVA to form antibacterial nanocomposite films. Bhowmick and Koul [20] assessed the activity of antimicrobial PVA-silver nanocomposite hydrogel films as wound dressing scaffolds. The films were formed using a freeze-thaw technique and showed sustained release of silver for 96 h. Zhang et al. [21] investigated the electrospinning of PVA in the presence of silver nitrate to create a PVA-silver nanowire composite with antibacterial properties against. against *E. coli* and *S. aureus*.

This paper outlines the novel, yet simple, inexpensive preparation of silver cross-linked PVA films that might provide a long-lasting protective hydrogel with a sustained release of anti-infective silver nanoparticles. The work characterizes PVA degradation, silver nanoparticle production, silver release, antibacterial activity and discusses how these films might be incorporated in a final dressing product for a burn survival kit intended for remote African villages.

## 2. Materials

Poly(vinyl alcohol) (88 mole% hydrolyzed, Mw ~125,000) was obtained from Polysciences, Inc (Warrington, PA, USA). Silver nitrate (>99.0%) was purchased from Sigma-Aldrich (St. Louis, MO, USA). Silver nanoparticle suspensions (20, 40 and 60 nm particle sizes) at a concentration of 0.02 mg/mL were also obtained from Sigma-Aldrich. All chemicals were used as supplied and without further purification. Deionized water was used in the preparation of all experimental PVA-silver formulations.

Antibacterial tests involved use of *E. coli* C1214, *Staphylococcus Aureus*. Newman and *Pseudomonas Aeruginosa*. Luria Bertani (LB) broth was obtained from Fisher Scientific. The *S. aureus* and *P. aeruginosa* bacterial species carry a reporter plasmid containing the *lux* gene, which is only expressed in viable bacteria, allowing for the quantification of bacteria via luminescence measurements.

## 3. Methods

### 3.1. Film Preparation

A stock solution of PVA was prepared as a 10% *w*/*w* gel by slowly adding 20 g PVA powder to 180 mL of rapidly stirred deionized water, heating to 85–90 °C and continuing with magnetic stirring and heating for approximately two hours. When a clear gel had formed the vessel was removed from heating and stored at room temperature in the laboratory. The PVA solution was diluted four-fold to 2.5% prior to use by adding 75 mL of water to 25 mL of the 10% PVA solution. A stock silver nitrate solution was then prepared in deionized water at a concentration of 10 mg/mL and stored covered with aluminum foil in a dark cupboard until required. Solutions of PVA and silver nitrate were then mixed in defined volumes to give wt ratios (PVA:Silver nitrate)ranging from 99%:1% to 95%:5% in the dried film form. Films were manufactured using 10 or 5 mL of these solutions to allow 200 μm (10 mL) or 100 μm (5 mL) films to be cast in 60 × 15 mm disposable polystyrene Petri dishes (Sarstedt Inc., Montreal, PQ, Canada). For example: to manufacture 200 μm films, either 250 or 1250 μL of the 10% silver nitrate solution was added to 10 mL of the 2.5% PVA solution to give final PVA:silver ratios of 99%:1% or 95%:5% respectively. Film thickness was measured using an electronic micrometer. Formulations chosen on the basis of an initial screening contained 1%, 3% or 5% *w*/*w* silver nitrate based on PVA content. The PVA-silver nitrate solutions in Petri dishes were loosely covered with aluminum foil and left overnight under ambient conditions in order for water to evaporate. As a next step, the Petri dishes containing the cast films were placed in an oven set at 65 °C for a further overnight period. Finally, the dried films were removed from the Petri dishes, placed on aluminum foil and heat cured in a second oven set at temperatures ranging from 80 to 140 °C for 90 min. For some experiments, films were heat cured for different times at 140 °C for 15, 30, 45 or 90 min. The films dried at 65 °C or the dried films which had then been heat cured were stored in a dark cupboard before evaluation. The final heat curing process distorted the films somewhat but it is anticipated that if the films were produced commercially, they would not be removed from the casting chamber until all heat curing was complete.

### 3.2. Film Swelling Studies

PVA films containing silver nitrate at 1%, 3% or 5% *w*/*w* (silver nitrate/PVA), with no curing or curing at 80 °C, 110 °C or 140 °C and with thicknesses of 100 μm (60 mg) or 200 μm (120 mg) were prepared as described above. These films were then stored for one week in the dark before use. Films with approximate diameters of 2 cm were then placed on moistened 0.2 μm filter discs (Millipore, Billerica, MA, USA) and weighed. The films and filters were covered with a layer of 0.5 mL of deionized water. After set time periods the filter discs and adherent PVA-silver gel were moved to a Millipore vacuum apparatus and vacuum was applied to draw all excess water from the filter over approximately 15 s. The combined PVA gel and filter were reweighed and then covered with a layer of excess deionized water (approximately 0.5 mL). The weight gain (swelling) and weight loss (dissolution) were then calculated as a percentage of the original dry film weight.

### 3.3. Silver Release Studies and Characterization

Films (10 mg squares cut from full film, n = 4) containing 1%, 3% or 5% *w*/*w* silver nitrate that had been heated for 90 min at 140 °C were placed in deionized water (5 mL) and all the media was sampled at regular intervals for silver analysis by atomic absorption (AA) spectroscopy. After the media was removed at each time point, 5 mL of fresh water was added to the film. The silver content in the release media was analyzed using a Perkin Elmer AA 560 spectrometer (Waltham, MA, USA) equipped with a Perkin Elmer silver lamp. This method allows the detection of low nanogram levels of silver. Initially a silver electrode system was used but the limit of detection was 10 µg/mL and unsuitable. Water was used as the release media because any buffer salts in the media would interfere with the AA methods. In order to determine the effect of temperature, a similar study was run on films containing 5% *w/w* silver nitrate, which had been heat cured at either 80, 110, 125 or 140 °C for 90 min. In a third investigation, the effect of curing time was determined through monitoring silver release from films heat cured at 140 °C for 15, 30, 45 or 90 min. Each release study was run for at least two weeks and the results plotted as the calculated percent silver released as a function of time.

The release media from selected PVA-silver films were subjected to particle size analysis using a Zetasizer Nano-ZS instrument (Malvern Instruments Ltd., Malvern, UK).

### 3.4. X-ray Diffraction of PVA-Silver Films

X-ray diffraction (XRD) experiments involved use of a Bruker Apex DUO instrument (Billerica, MA, USA) equipped with a Cu Kα source (λ = 0.15418 nm) and a heating stage. In order to learn more about the PVA crosslinking process, an XRD experiment was run on a dried PVA film containing 5% *w*/*w* silver nitrate with in situ heating of the film at 140 °C in the XRD instrument. X-ray diffractograms were collected once the sample film had reached the set temperature and thereafter at 15-min intervals for two hours. The resulting diffractograms were then plotted as an overlay against 2θ, the Bragg diffraction angle.

### 3.5. Scanning Electron Microscopy (SEM)

SEM images of PVA-silver films were obtained using an FEI Helios 650 Nanolab dual beam instrument (Hillsboro, OR, USA) and imaging at low voltage with secondary electrons (1 keV for true surface imaging). For the SEM experiments, the films were mounted on aluminum stubs using carbon adhesive tape. The edges of the films were coated with colloidal silver in order to make an electrical connection between the film and the aluminum stub. The film area of interest was not painted with colloidal silver but was coated with a 5 nm-thick electrically conductive layer of carbon using a Leica EM MED020 modular coating system (Leica Microsystems Inc., Concord, ON, Canada).

### 3.6. Antimicrobial Activity of Silver-Loaded PVA Films

In order to evaluate the antimicrobial activity of PVA-silver films, we selected three film sets loaded with 1%, 3% or 5% *w*/*w* silver nitrate, each of which had been previously heat cured at 140 °C for 90 min. The weight of each film sample was 28–30 mg and eight replicates were prepared for each film type. Controls, also prepared as eight replicates, consisted of PVA powder (27 mg) or 10% *w*/*w* PVA gel (270 mg), each of which was added to 2.5 mL of water. Prior to the experiments, each bacterial species was cultured overnight from freezer stocks (10 μL bacterial sample into 10 mL Luria Bertani (LB) broth) followed by a second sub-culture (100 μL in 10 mL LB broth) until an optical density of ~0.3–0.5 at λ = 600 nm was reached, as measured using an Eppendorf BioPhotometer (Eppendorf Canada, Mississauga, ON, Canada), at which point bacteria were used as outlined below. 

The antibacterial activity of PVA-silver films was assessed by placing pre-weighed film samples in nutrient media containing uropathogenic *E. coli*, *S. aureus* or *P. aeruginosa* (5.00 × 10^5^ CFU/mL) with 10 mL of 100% culture media per bottle. The sample bottles were incubated at 37 °C with shaking at 100 rpm for 48 h. Aliquots were taken at 0, 6, 24 and 48 h after inoculation and bacterial numbers were determined as colony-forming units (CFU), as well as by luminescence measurements for *S. aureus* and *P. aeruginosa*, at each time point. Luminescence measurements were conducted using a Tecan Infinite M200 Pro Luminometer (Tecan Group Ltd., Männedorf, Switzerland).

### 3.7. Antimicrobial Efficacy of Aqueous Release Media from PVA-Silver Films

The antimicrobial activity of various release media was determined using 96-well plate assays against *S. aureus* and *P. aeruginosa* but not against *E. coli* because tests using PVA-silver films showed that antimicrobial activity was not strong in that case. In each test, 40 μL of LB broth was aliquoted into each well, to which 60 μL of LB broth containing 8.33 × 10^5^ CFU/mL of bacteria was added to give a final bacterial concentration of approximately 5.00 × 10^5^ CFU/mL. The exact starting concentration was verified via CFU counts. Control wells were set up as follows: the bacterial control contained 60 μL of culture in 40 uL of sterile double-distilled water and the negative bacterial control contained 100 μL of sterile double-distilled water only. Following gentle mixing, each 96-well plate was incubated for four hours at 37 °C and 120 rpm. At 0, 1, 2, 3, and 4-h time points the luminescence was measured to determine bacterial viability. In addition to luminescence, viable bacteria were also quantified via CFU counts at the 4-h time point to ensure correlation with the luminescence measurements.

Aqueous release media from PVA-silver films were obtained at daily intervals up to 10 days. After each sampling, the liquid removed was replaced with an equal volume of fresh nutrient broth. Four 5 mL replicates of each release medium were used for this study. In each case half of the release medium (2.5 mL) was added to bacteria-containing nutrient broth at 33% concentration. Incubation and sampling were then carried out as for the experiments with the controls. 

## 4. Results and Discussion

### 4.1. Appearance of PVA-Silver Films

The probable mechanism of PVA crosslinking using silver and heat is shown in Figure 1. The visual appearance of 5 cm-diameter PVA-silver films after drying and before heat curing, as well as films containing 1%, 3%, or 5% *w*/*w* silver nitrate which had been heat cured at various temperatures for 90 min, is shown in Figure 2. All films became light brown in color after overnight drying at 65 °C and the color of the films became increasingly darker as the heat-curing temperature increased. 

The appearance of the same films after immersion in water and constant agitation for one week at 37 °C in an incubator shaker is shown in Figure 3. The films that had only been dried and the films dried and heat cured at 80 °C for 90 min dissolved rapidly in water; however, the films heat cured at 110 °C for 90 min were only partly dissolved after one week and the films heat cured at 140 °C for 90 min appeared to remain intact. The aqueous leachate from the films took on a yellow colour, which was particularly noticeable for the films that showed some degree of dissolution in water.

The observed change in the color of PVA films containing silver nitrate is consistent with the reduction of silver ions to silver metal nanoparticles. In essence, PVA has a number of active hydroxyl groups available for binding silver. The heat-induced reaction of silver with PVA to generate silver nanoparticles in situ can be described by the equations below.
>R−OH + Ag^+^ → >R−O−Ag + H^+^
>R−O−Ag → −R=O + Ag^0^
>R−OH + Ag^+^ → −R=O + Ag^0^ + H^+^

In these equations, −R=O represents a monomer of partially oxidized PVA at the reaction surface while H^+^ represents the nitric acid by-product.

As noted by Longenberger and Mills [22] and more recently by Porel et al. [23], on heating PVA in the presence of silver nitrate the polymer can act as a reducing agent, a stabilizer and as a matrix for homogeneous distribution and immobilization of silver. It is possible that a side effect of silver nanoparticle production might be the generation of nitrate ions with unknown effects in wounds. However, since silver nitrate solution is available by prescription to treat burns and wounds it is unlikely that such effects might cause any local toxicity.

The release media from selected PVA-silver films were examined using particle size analysis. Khanna et al. [24] reviewed the synthesis and characterization of PVA-silver nanocomposites by chemical reduction methods, specifically using hydrazine hydrate or sodium formaldehyde sulfoxylate (SFS), which resulted in silver nanoparticles with size less than 10 nm as determined using transmission electron microscopy (TEM). As also pointed out by these authors, there are reports in the literature of the synthesis of silver nanoparticles protected by PVA [23,24]. Our particle size data shown in Figure 4, when compared with similar data for silver nanoparticle standards, indicated that the release media have two populations; first, a sub-100 nm set and, second, a group of much larger particles, some of which are a micron or more in dimension. The sizes likely reflect the nanoparticle size presented to potential bacteria in a wound or burn setting but do not necessarily report the size within the PVA films as the particles have been exposed to water for some time.

### 4.2. Film Swelling

After an initial brief swelling period of about 10 min, 100 μm- and 200 μm-thick films that were uncured or cured at 80 °C were fully dissolved within 2.5 h regardless of silver nitrate loading. (Figure 5A,B). A similar swelling profile was observed for films cured at 110 °C, except that the films dissolved more slowly than the films cured at 80 °C (Figure 5C,D). For example, after curing at 110 °C for 2.5 h, the 200 μm-thick films containing 1% silver nitrate or 5% silver nitrate remained approximately 60% and 300% swollen respectively (Figure 5D). Furthermore, for films cured at 110 °C the degree of swelling was larger when these films were loaded with 5% silver nitrate (only) and when the films were thicker (200 μm). The 5% silver nitrate/110 °C/200 μm-thick films failed to fully dissolve and remained 200% swollen after five hours and 50% swollen after 24 h. Curing at 140 °C in the case of both 3% and 5% silver loadings resulted in films that swelled slowly after 2.5 h by approximately 1400% and 2600% in 100 and 200 μm-thick films respectively (Figure 5E,F). These films remained very swollen and had only dissolved to a ~600% swollen level after 100 h, remaining swollen to almost the same extent after 250 h. However, films cured at 140 °C with 1% silver loading swelled quickly and dissolved to approximately the original dry weight after four hours.

When PVA films are exposed to water, the films initially swell as a result of water penetration into the polymer. However, when films containing silver, added as silver nitrate, are heat cured under relatively mild conditions at 80 °C (e.g., as in Figure 5A,B), this initial period of swelling is rapidly followed by dissolution in water. The same process appears to occur at 110 °C, albeit at a slower rate of dissolution (Figure 5C,D). When heating at 140 °C, the PVA film is largely insoluble and, despite some dissolution, as indicated in Figure 5E,F, remains in the swollen state. In general, one would expect an increase in PVA cross-linking to restrict film swelling, as might occur at even higher silver nitrate loadings, but this possibility was not pursued in our research. The outcome of the heat-curing process at 140 °C, in the presence of 3% or 5% *w*/*w* silver, is a silver cross-linked film retaining some solubility in water and which allows silver release. As suggested in other PVA cross-linking systems, a reduction in macromolecular mesh size may also take place and limit silver ion or silver nanoparticle diffusion into the surrounding medium. For the purposes of this project, an ideal film swelling was not defined; however, the possibility to retain a silver-containing PVA film in an intact form while exhibiting swelling and silver release was considered to be a suitable goal. In vivo, it is possible that hydrogel-forming films that reside on wounds for more than a few days may be poorly oxygenated with the risk of anaerobic bacteria infection. In the rare cases in Africa where the hydrogel is present for more than three days, the continued protection of the wound with the associated inhibition of inappropriate wound healing (and subsequent scarring or deformation) is considered more important that the risk of such infections.

### 4.3. Release of Silver from PVA-Silver Films

The in vitro release of silver from PVA-silver films heat cured at 140 °C for 90 min as a function of silver loading is shown in Figure 6 as percent silver released over time for up to two weeks. As demonstrated, the quantity of silver released in the initial phase decreased with increasing concentration of silver in the film. For example, after five days the percent silver released from films containing 1%, 3% or 5% *w*/*w* silver nitrate was 43, 22 and 17.5% respectively. Between nine and 14 days, the rate of silver release from each of the three film types was roughly equal and, after 28 days, release of silver had leveled out at 69%, 41.5% and 31.8% respectively. These findings are consistent with visual observations in that films with 1% and 3% *w*/*w* silver nitrate exhibit rapid, partial or full dissolution when immersed in water. When 5% *w*/*w* silver nitrate is present, PVA films remain largely intact and, although a certain percentage of each film does dissolve in water, the burst release phase of silver ions or nanoparticles is modulated.

The in vitro release of silver from PVA-silver films as a function of heat-curing temperature is illustrated in Figure 7, in which percent silver released is plotted against time for films containing 5% *w*/*w* silver nitrate, which have been heated for 90 min at 80, 110, 125 or 140 °C. As illustrated, films heated at 80 or 110 °C released about 90% of the silver content very rapidly. When films were heated at 125 °C there was also a rapid initial release of ~70% of the silver content. However, when films were heat cured at 140 °C, there was an initial release of only ~20% of the silver content and then a gradual release of silver over the following two weeks.

The in vitro release of silver from PVA-silver films as a function of heat-curing time when films containing 5% *w*/*w* silver nitrate were heated at 140 °C is depicted in Figure 8, in which percent of silver released is plotted against time. The plots follow a similar pattern to that seen in Figure 6 and Figure 7, in that the amount of silver initially released decreases as heat-curing time increases, but after about five days the rate of silver release is approximately equal in each case. The final percent silver released after 14 days was 75, 60, 25 and 20% for films heated for 15, 30, 45 or 90 min respectively.

The release curves depicted in Figure 6, Figure 7 and Figure 8 point to PVA-silver films containing 5% *w*/*w* silver nitrate and heat cured at 140 °C for 90 min as presenting a potentially optimum combination of insolubilized PVA, a reservoir of bound silver in the insoluble PVA film and a gradual release of silver nanoparticles. 

### 4.4. XRD and SEM Studies of PVA-Silver Films

The changes in the XRD pattern of PVA films containing 5% *w*/*w* silver nitrate as a function of holding time in the instrument at 140 °C are presented in Figure 9. As shown, the XRD peaks at 2*θ* = 38.2° and 44.4°, characteristic of the 111 and 200 planes respectively in the cubic crystal structure of silver metal, increased in intensity with curing time. As discussed in the literature, these peaks provide further evidence that silver nanoparticles are created in the films during the heat curing process and may subsequently be released on exposure to aqueous media [25,26,27,28,29,30]. 

The SEM image of a PVA-silver film containing 3% *w*/*w* silver nitrate and heat cured at 140 °C for 90 min is shown in Figure 10, providing further confirmation of the presence of silver nanoparticles in the heat-cured PVA films. The mean particle size as determined using Image J software is 20.8 nm in this case.

There are numerous reports of the manufacture of silver nanoparticle loaded composite films for use in antimicrobial applications [26,27,28,29,30]. All these studies used silver nitrate as the starting material and X ray diffraction and electron microscopy to establish the presence of elemental silver as nanoparticles in the films and to comment on nanoparticle size. The films were manufactured from biocompatible materials such as gelatin [26], cellulose [27], chitosan [28]. agar [29] or poly lactic acid [30] and antimicrobial activity was demonstrated in vitro only using Gram-positive (*S. aureus*) and Gram-negative bacteria (*P. aeruginosa* or *E. coli*) similar to the studies reported here.

### 4.5. Antimicrobial Activity of Silver-Loaded PVA Films

The PVA-silver films containing three different concentrations of silver showed antimicrobial effects when incubated with bacteria (Figure 11). However, the 1% *w*/*w* silver nitrate film was not potent enough to inhibit the growth of *E. coli* (Figure 11A). The same film was bacteriostatic towards both *S. aureus* and *P. aeruginosa*, with the effect more prominent for the latter bacterium (Figure 11B,C). Both 3% and 5% *w*/*w* PVA-silver films significantly decreased the growth of all three bacteria for up to four hours (3% film against *E. coli* and *S. aureus*, 5% film against *P. aeruginosa* and *S. aureus*) or 24 h (3% film against *P. aeruginosa* and 5% film against *E. coli*). After these time points, bacterial growth accelerated; however, the bacterial concentration was still significantly lower than that for the controls after 48 h.

### 4.6. Antimicrobial Activity of Aqueous Release Media from Silver—Loaded PVA Films

Films with different concentrations of impregnated silver nitrate were incubated in water for up to 10 days, with the supernatant, which contained any released silver, collected and replaced with fresh water every 24 h, with the exception of days 8–10, which were sampled as one solution without replacing water in between. When the collected supernatants were tested for antimicrobial activity, supernatants collected from 1%, 3%, and 5% *w*/*w* silver nitrate films only showed differences in bacterial inhibition when collected up to 3-days post-incubation for *P. aeruginosa* (Figure 12A), and up to only 1-day post-incubation for *S. aureus* (Figure 12B). Once past those time points, the collected supernatants showed similar antimicrobial activity regardless of the silver concentration in the film. However, all of the collected supernatants showed significant antimicrobial activity against *P. aeruginosa* when compared to controls, which grew to approximately 3.73 × 10^7^ CFU/mL. Supernatants collected at later dates were less potent against *S. aureus* when compared to the controls, which had approximately 7.00 × 10^7^ CFU/mL of bacteria. For Figure 12A,B, data from the controls were not included for better comparison between supernatants collected from the various samples.

Whilst these studies used bacteria growing in planktonic conditions it is possible that in wounds and burns, left for extended periods of more than a few days, that the bacteria may form biofilms which may prove to be a bigger antimicrobial challenge for PVA-silver films. Additionally, it is possible that some silver-resistant bacteria might be encountered, again providing a more significant challenge. However, in Western hospitals silver based wound dressings are generally not intended for use over long time periods and it is assumed that, similarly, in African countries the extended use of these PVA films would be avoided. If extended residence times of PVA hydrogel films on wounds and burns was required it is hoped that the initial inhibition of bacterial growth and the hydrogel protection of the wound would still provide a better outcome for the patient. Previously, using antibiotic loaded PLGA nanofibers we developed a uniform animal orthopedic model for in vivo testing [31]. However, it is very difficult to create a uniform infected burn wound model. We felt it unnecessary to develop an anti-infective animal model because similar materials are already used in humans and controlled bacterial testing in vitro with both gram negative and positive bacteria establish that release profiles allow strong antibacterial effects in the presence of huge loads of bacteria.

## 5. Conclusions

The use of silver as an antimicrobial additive in commercially available wound dressings is now well established in western countries and, amongst these dressings, there are products such as Acticoat™ 7 and Acticoat™ Moisture Control, which contain nanocrystalline silver and rely on silver nanoparticles for their anti-infective activity. As discussed in this paper, we have discovered that by subjecting PVA films containing silver nitrate to a sequential oven-drying and heat curing treatment, it is possible to generate crosslinked PVA films with in situ creation of silver nanoparticles which exhibit delayed release properties on exposure to water. Further, in the context of a burn wound dressing application, the use of 5% *w*/*w* loadings of silver nitrate combined with heat treatment at 140 °C for 90 min, may be optimal in terms of both insolubilizing PVA and delaying silver release. The presence of silver nanoparticles in the PVA films and release media and the potent silver based antimicrobial activity have been demonstrated in this study. We have therefore concluded that silver-cross-linked PVA film may be a suitable material as a basis for a simple and inexpensive burn wound dressing.

## Figures and Tables

**Figure 1 nanomaterials-11-00096-f001:**
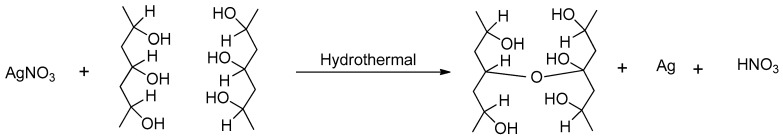
Cross-linking of poly(vinyl alcohol) (PVA) with silver nitrate under hydrothermal conditions.

**Figure 2 nanomaterials-11-00096-f002:**
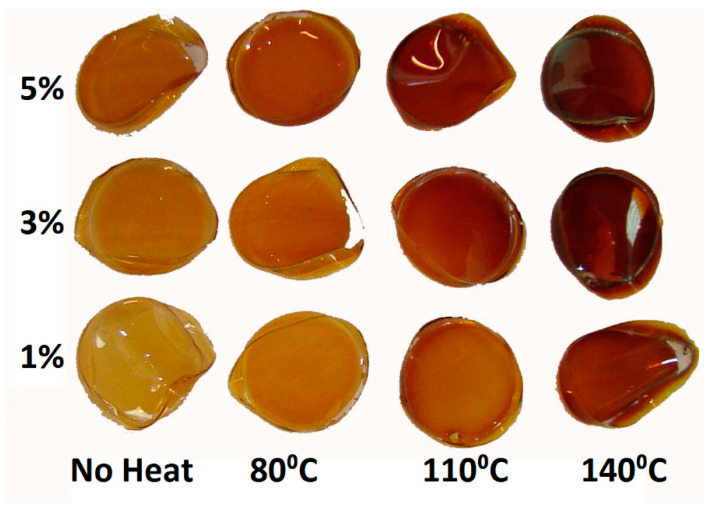
Silver cross-linked PVA films containing 1%, 3% or 5% *w*/*w* silver nitrate, dried at 60 °C overnight and then heat cured at various temperatures for 90 min.

**Figure 3 nanomaterials-11-00096-f003:**
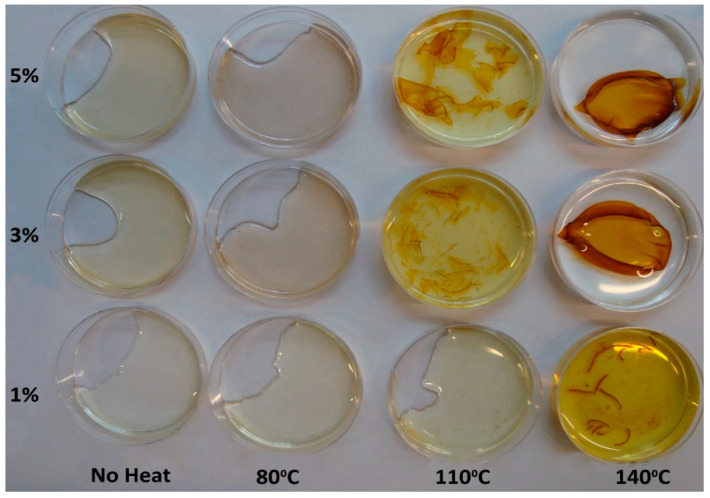
The same films as shown in Figure 2 after seven days’ incubation in water at 37 °C with constant agitation at 70 rpm.

**Figure 4 nanomaterials-11-00096-f004:**
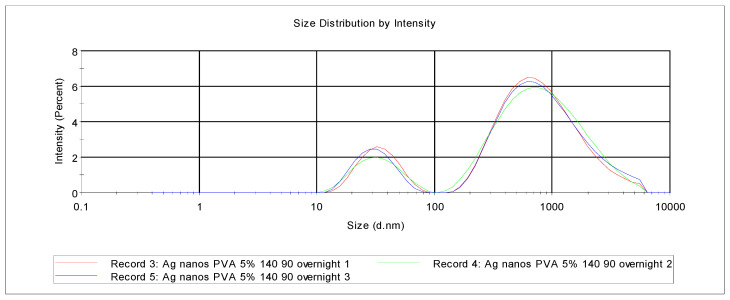
Particle size analysis of the aqueous release medium after immersing PVA/silver film (5% *w*/*w* silver nitrate, heat cured at 140 °C for 90 min) in deionized water overnight.

**Figure 5 nanomaterials-11-00096-f005:**
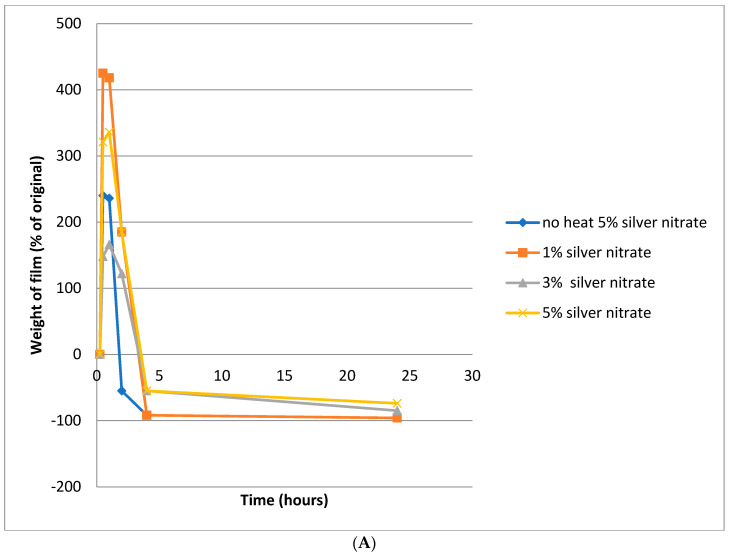
(**A**) Degradation of 100 µm PVA/silver films heated at 80 °C for 90 m and then immersed in water; (**B**) degradation of 200 µm PVA/silver films heated at 80 °C for 90 m and then immersed in water; (**C**) degradation of 100 µm PVA films heated at 110 °C for 90 m and then immersed in water; (**D**) degradation of 200 µm PVA/silver films heated at 110 °C for 90 m and then immersed in water; (**E**) degradation of 100 µm PVA films heated at 140 °C for 90 m and then immersed in water; (**F**) degradation of 200 µm PVA films heated at 140 °C for 90 m and then immersed in water.

**Figure 6 nanomaterials-11-00096-f006:**
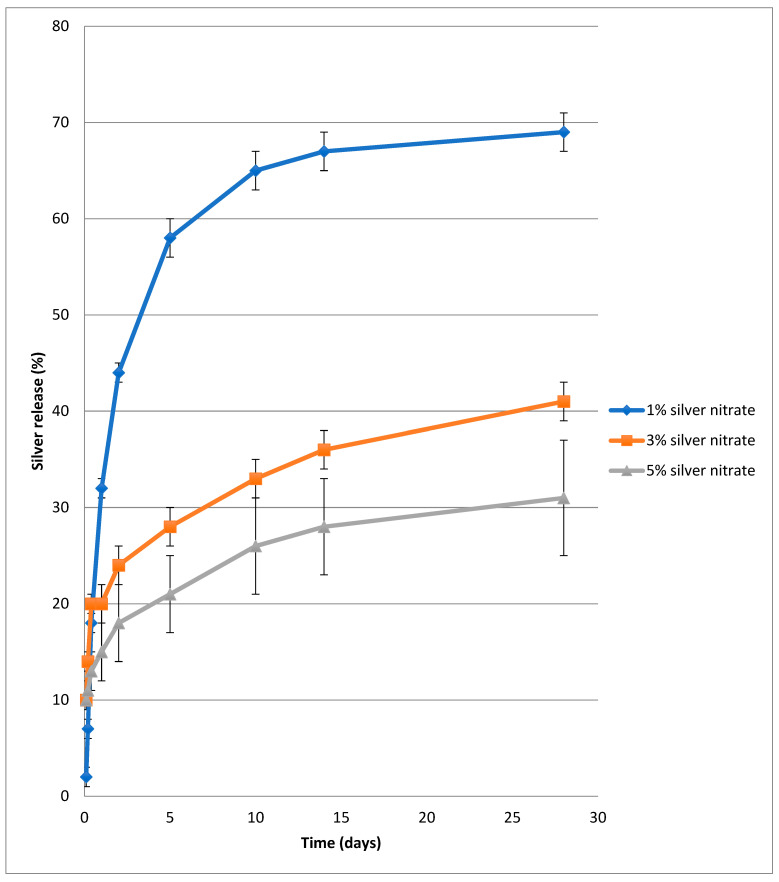
In vitro release of silver from PVA/silver nitrate films heated at 140 °C for 90 m and then immersed in water as a function of silver content.

**Figure 7 nanomaterials-11-00096-f007:**
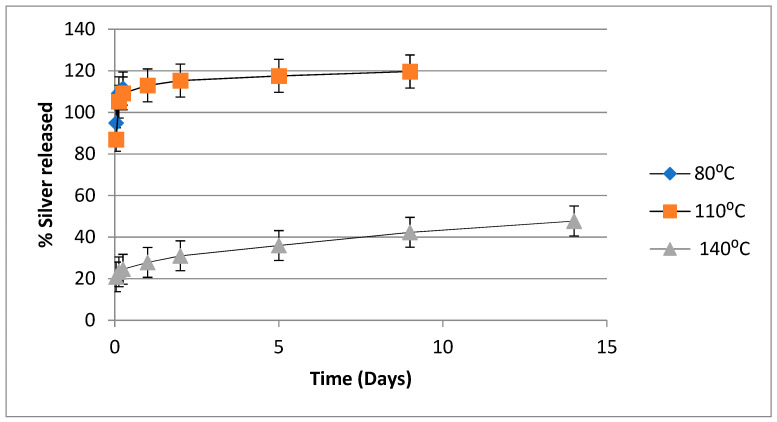
In vitro release of silver from PVA/silver nitrate films in water as a function of temperature after films with 5% silver nitrate loading have been heated at 80, 110 and 140 °C for 90 min.

**Figure 8 nanomaterials-11-00096-f008:**
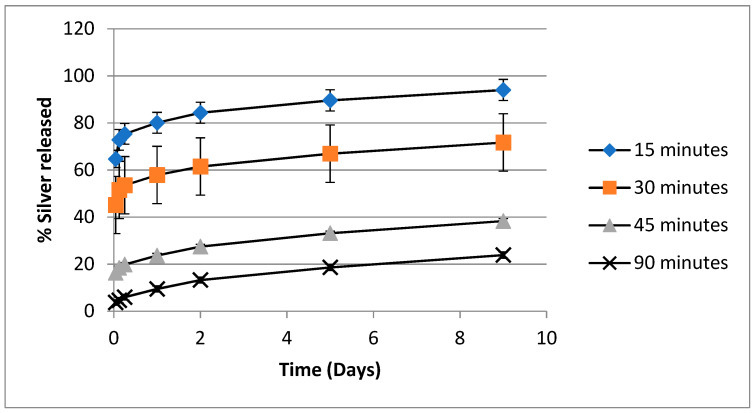
In vitro release of silver from PVA/silver nitrate films in water as a function of heating time after films with 5% silver nitrate loading have been heated at 140 °C.

**Figure 9 nanomaterials-11-00096-f009:**
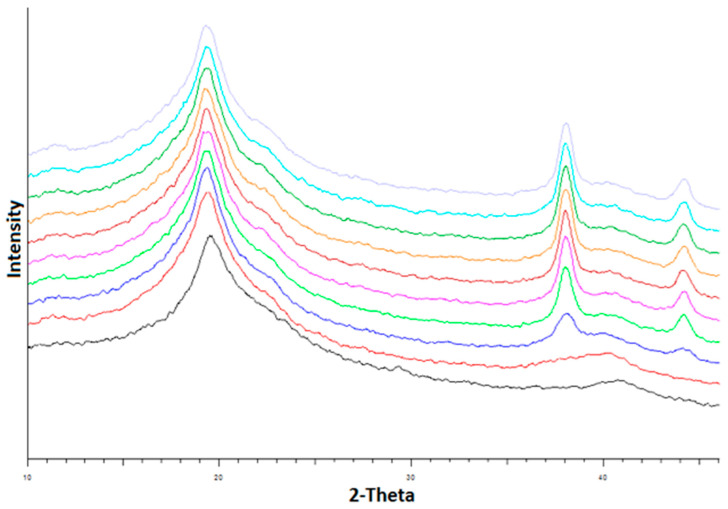
X-ray diffractograms for a PVA film containing 5% *w*/*w* silver nitrate as a function of heating time at 140 °C. From bottom to top, the curves show the film diffractograms before heat curing, after the instrument temperature had reached 140 °C and at 15 min intervals thereafter. The increasing peaks at 38° and 44.5° 2-Theta arise from the creation of silver nanoparticles over time.

**Figure 10 nanomaterials-11-00096-f010:**
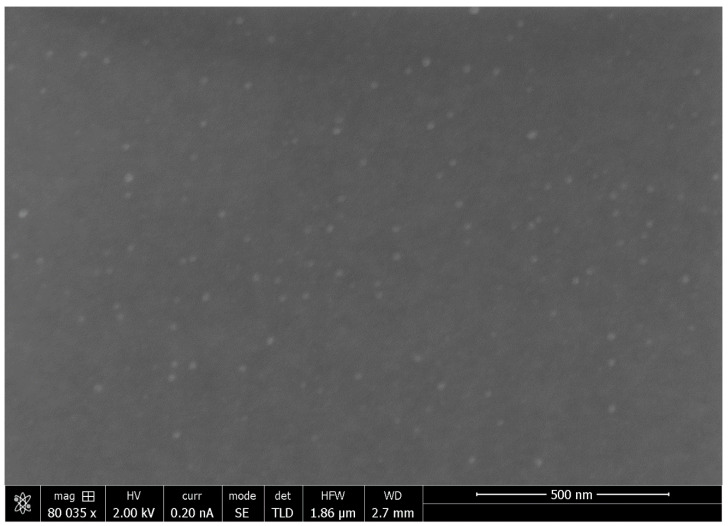
SEM image of a silver-loaded PVA film containing 3% *w*/*w* silver nitrate that had been heat cured for 90 min at 140 °C.

**Figure 11 nanomaterials-11-00096-f011:**
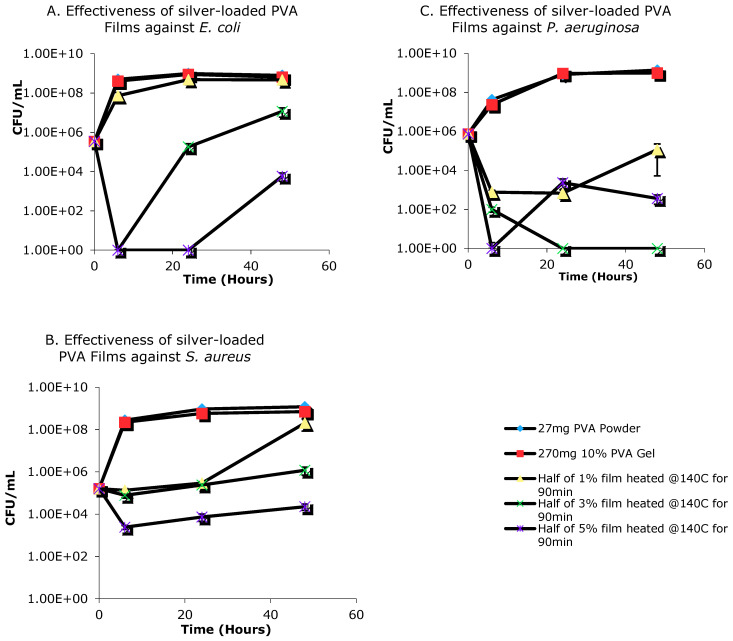
Evaluation of in vitro antimicrobial activity of PVA-silver films against (**A**) *E. coli*; (**B**) *S. Aureus* and (**C**) *P. Aeruginosa*.

**Figure 12 nanomaterials-11-00096-f012:**
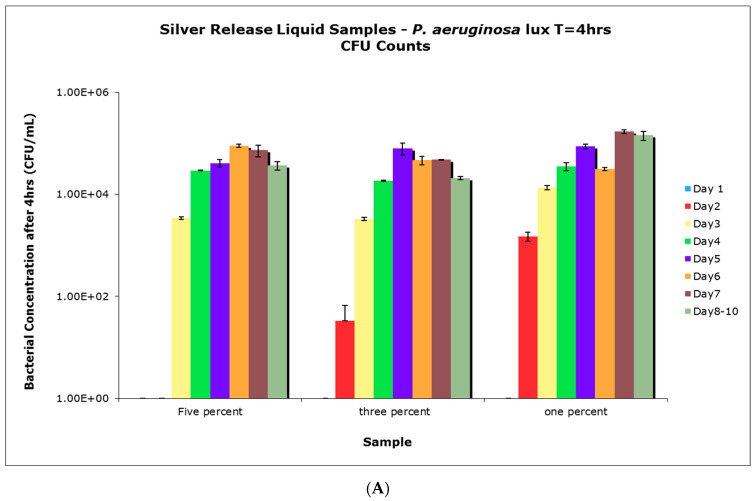
(**A**) Colony-forming units (CFU) count results from antimicrobial tests (*P. aeruginosa*) using supernatants collected from incubation of PVA/silver films over a 10-day period; (**B**) CFU count results from antimicrobial tests (*S. aureus*) using supernatants collected from incubation of PVA/silver films over a 10-day period.

## Data Availability

Data sharing not applicable.

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
