# Peer review of "The Design, Characterization and Antibacterial Activity of Heat and Silver Crosslinked Poly(Vinyl Alcohol) Hydrogel Forming Dressings Containing Silver Nanoparticles"

_nanomaterials, 2021, doi:10.3390/nano11010096_

Round 1
Reviewer 1 Report
I have carefully read the paper The design and characterization of a silver nanoparticle loaded, hydrogel forming wound dressing material and I believe there are many critical points.
The key words mucoadhesion, oral dosage forms, regulatory and cosmetic products are not appropriate for this study.
The topic of the paper is interesting but is not sufficiently supported by experimental data, some experiments have been omitted or not carried out correctly
The introduction is too long, it contains numerous superfluous details and 27 bibliographic references are excessive. This introduction should be shortened and must report only the aspects necessary to present the work. The figure must be eliminated and only the bibliographic references functional to the work must be reported.
The authors state that the preparation method allows the formation of nanoparticles but support this statement only by bibliographical references, there are no characterization studies of the formed nanoparticles. The SEM analysis figure is not clear and does not allow to identify the silver nanoparticles.
Moreover this method of preparation generates nitrate ions. Did the authors consider the toxicity of nitrate? Nitrate ions are not suitable for application on wounds because they could be toxic to tissues.
These films have been designed for a long time of application and this could allow the development of anaerobic bacteria. Authors should perform oxygen perspiration tests to be sure the wound is oxygenated during this long period.
Pagg.4-5. The method of preparation of the films is not detailed and precise, as required in a scientific work (e.g. sufficient quantity, various period of time ...).
Pag. 5. “Film swelling studies”: how was the thickness of the films measured?
Pag. 5. “silver release studies and characterization”: the release study was not approached properly. Pharmacopoeias (see USP, Eur Ph, …) indicate suitable instrumentation and methods that have not been considered. Furthermore, it is necessary to use a suitable dissolution medium, not simply water.
Pag. 6. The term “antimicrobial efficacy is uncorrected plese use “antimicrobial activity.
Pag. 6, line 9: mgs is uncorrected.
Pag. 7: the films presented in figure 2 are very irregular and with shape (the diameter refers to a regular shape) and thickness not uniform. The preparation technique needs to be revised.
Pag 8: the formation of nanoparticles must be studied, it cannot be confirmed by bibliographic data. The partcle size must be studied by a specific method (DLS). It is not clear how the size of the nanoparticles was determined.
Pag. 9, line19-20. The explanation for the film swelling is obvious, but scientific data are necessary to describe and discuss the swelling phenomenon.
The conclusions are very general and are not supported by correct and precise scientific data.
Author Response
We thank the reviewer for the detailed and very pertinent points. We have responded in a point by point manner below.
I have carefully read the paper The design and characterization of a silver nanoparticle loaded, hydrogel forming wound dressing material and I believe there are many critical points.
The key words mucoadhesion, oral dosage forms, regulatory and cosmetic products are not appropriate for this study.
Response: we agree. We are not sure how these keywords appeared. We have changed them to : wound dressing, silver nanoparticles, PVA hydrogel, anti-infective.
The topic of the paper is interesting but is not sufficiently supported by experimental data, some experiments have been omitted or not carried out correctly
The introduction is too long, it contains numerous superfluous details and 27 bibliographic references are excessive. This introduction should be shortened and must report only the aspects necessary to present the work. The figure must be eliminated and only the bibliographic references functional to the work must be reported.
Response: We have rewritten and improved the Introduction. We have removed a big section that was unnecessarily overloaded with references and removed 8 references.
The authors state that the preparation method allows the formation of nanoparticles but support this statement only by bibliographical references, there are no characterization studies of the formed nanoparticles. The SEM analysis figure is not clear and does not allow to identify the silver nanoparticles.
Response: Because the nanoparticles are embedded in the PVA film it was difficult to characterize them apart from using SEM ( which we agree is not optimal). However we did include X ray data in Figure 9 showing the heat dependent creation of elemental silver supporting the SEM data.
Moreover this method of preparation generates nitrate ions. Did the authors consider the toxicity of nitrate? Nitrate ions are not suitable for application on wounds because they could be toxic to tissues.
Response: We agree that the generation of nitrate ions is not optimal. However the amount is very small so that in the largest film with the highest silver nitrate loading (120 mg film with 5% ) the total weight of nitrate present is only 2 mg. We note that topical silver nitrate solution is available by prescription in a 0.5% to 50% solution for application to wounds and burns so we feel this small amount of unaccounted for nitrate is unlikely to be a problem. We have now discussed this in the Results and Discussion section.
These films have been designed for a long time of application and this could allow the development of anaerobic bacteria. Authors should perform oxygen perspiration tests to be sure the wound is oxygenated during this long period.
Response: The reviewer raises an excellent point. Usually hydrogel films like these would be applied for much shorter periods. However, in third world areas (especially Africa) there may be a considerable delay in reaching medical care for patients (especially children). One of our authors (Robin Evans- a plastic surgeon) visited Africa many times during the time course of this project to assess the needs of patients. Although our first films degraded more rapidly, Robin told us to extend the time to meet all scenarios. The point being that hydrogel protection of the wound helps to prevent poor wound healing and scarring and makes the job of the clinician much simpler. The possible negative side effects of a lengthy residence PVA hydrogel (poor oxygenation and possible anaerobic bacteria) were considered less problematic than leaving wounds prematurely exposed from a more rapidly degrading film. In the majority of cases the films would be removed within a few days. We have now addressed this point in the Discussion.
Pagg.4-5. The method of preparation of the films is not detailed and precise, as required in a scientific work (e.g. sufficient quantity, various period of time ...).
Response: Yes, we now see the lack of detail in this section. We have rewritten the methods as requested.
Pag. 5. “Film swelling studies”: how was the thickness of the films measured?
Response: We used a digital micrometer and this information has now been included in the Methods.
Pag. 5. “silver release studies and characterization”: the release study was not approached properly. Pharmacopoeias (see USP, Eur Ph, …) indicate suitable instrumentation and methods that have not been considered. Furthermore, it is necessary to use a suitable dissolution medium, not simply water.
Response: The methods section was written poorly- our apologies. We have rewritten this section in detail. This Atomic Absorption method allows the detection of low nanogram levels of silver. Initially a silver electrode system was used but the limit of detection was 10ug/ml and unsuitable. Water was used as the release media because any buffer salts in media would interfer with the AA methods.
Pag. 6. The term “antimicrobial efficacy is uncorrected plese use “antimicrobial activity.
Response: We have corrected this.
Pag. 6, line 9: mgs is uncorrected.
Response: Corrected
Pag. 7: the films presented in figure 2 are very irregular and with shape (the diameter refers to a regular shape) and thickness not uniform. The preparation technique needs to be revised.
Response: When these films were initially formed in the petri dish they were flat and unwrinkled and stayed that way when lifted off the petri dish. However subsequent controlled heating methods distorted the films. In these proof of principle experiments we did not think this important. If these films were to be manufactured commercially, then they would not be removed from the drying/heating container until the end of the process. We have now commented on this in the Discussion.
Pag 8: the formation of nanoparticles must be studied, it cannot be confirmed by bibliographic data. The partcle size must be studied by a specific method (DLS). It is not clear how the size of the nanoparticles was determined.
Response: It is very difficult to examine the nanoparticles when they are embedded within the PVA films. Evidence of nanoparticles is provided (rather poorly) by SEM photography but also by Xray diffraction (Figure 9) which shows the heat dependent generation of elemental silver witnessed by the increasing silver peak size. When the release studies were performed, the media was analysed by laser diffraction size analysis to report particle sizes of sub 100nm with larger aggregates. However, these sizes do not necessarily report the true particle size in the dried film as the particles have been exposed to water for some time. However these sizes may be informative to clinicians who might be interested in the possible size of silver nanoparticles presented to the bacteria. We have included these comments in the new Discussion section.
Pag. 9, line19-20. The explanation for the film swelling is obvious, but scientific data are necessary to describe and discuss the swelling phenomenon.
Response. These films are very thin and hydrogel systems absorb water rapidly so it is very difficult to measure a moving (swelling) film boundary. For films that reach a stable swollen state it might be possible to examine the swelling using a stereo microscope or similar. But still the accurate measurement of a moving film boundary is difficult. We believe that a better more quantitative method is to simply measure the increased weight of the hydrated film using the assumption of swelling. This method of measuring the rate of swelling of hydrogel films has been published by our group previously but it seems unnecessary to include an additional reference at this time. We will happily do so if needed. ( see: Pharm Res. 2002 Apr;19(4):411-7. doi: 10.1023/a:1015175108183)
The conclusions are very general and are not supported by correct and precise scientific data.
Response. Upon reflection and considering the reviewer’s discussion about nanoparticle characterization we agree with the reviewer. The conclusion has been rewritten to remove the discussion about the characterization and nature of the nanoparticles.
Reviewer 2 Report
The methods used to produce the silver infused hydrogels were standard. The most important aspect of the study was that their method would provide an economic and readily available addition to burn treatment for use in underdeveloped nations. This is reason alone for the publication of this paper. However there are some biological errors that the authors did not consider. First they tested the effectiveness of their silver hydrogel against bacteria growing in planktonic conditions. It is highly likely that the bacteria infecting a burn will be growing as biofilm. They also did not consider testing their silver hydrogel against silver resistant bacteria. The sil operon is carried on plasmids and widely distributed throughout nature, particularly in hospitals. Thus their work sort of represents the effectiveness of the hydrogel under the best of conditions, e.g. bacteria growing in planktonic form w/o silver resistance genes (unfortunately that is not the situation in nature). These sophistications should have been addressed in their discussion.
Author Response
We have addressed the important points made by the reviewer with this new text (below) at the end of the antimicrobial section of Results and Discussion.
"Whilst these studies used bacteria growing in planktonic conditions it is possible that in wounds and burns, left for extended periods of more than a few days, that the bacteria may form biofilms which may prove to be a bigger antimicrobial challenge for silver films. Also, it is possible that some silver-resistant bacteria might be encountered, again providing a more significant challenge. However, in western hospitals silver based wound dressings are generally not intended for use over long time periods and it is assumed that, similarly, in African countries the extended use of these PVA films would be avoided. If extended residence times of PVA hydrogel films on wounds and burns was required it is hoped that the initial inhibition of bacterial growth and the hydrogel protection of the wound would still provide a better outcome for the patient."
Reviewer 3 Report
This manuscript deals with the preparation of PVA hydrogel loaded by different amount of silver nanoparticles which act not only as anti-bacterial agents, but also allow to cross link the polymer chains.
The topic is interest, above all in the context of its application
Graphics and figures should be reconsidered for publication and not for presentation.
Author Response
We have rewritten much of the manuscript and moved Figure 1 and 2 appropriately. We feel the graphics illustrate to the reader the morphology of the various films and the degree of degradation in water. We feel that it would be difficult to simply describe these aspects in the text and we hope that the editor is Ok with publishing them. If not we will be happy to remove figures and graphics.
Round 2
Reviewer 1 Report
I carefully read both the new manuscript and the answers provided by the authors. The authors responded diplomatically but the study has not been improved in its fundamental parts and is not suitable for publication in this journal.
It is a scientific research and all aspects must be demonstrated with appropriate experiments, conducted correctly. For example x-rays are not suitable for determining nanoparticles (size) and if the authors have difficulty characterizing nanoparticles they cannot think of publishing the work in a journal that focuses on nanomaterials.
The title of the work is: The design and characterization of a silver nanoparticle loaded, hydrogel forming wound dressing material. The design is not innovative, the characterization of the nanoparticles has not been performed, the activity on the wounds has not been studied. Only a modest part is reported on anticrobial activity.
Some precise indications on the exact methods of carrying out the experiments were disregarded.
Author Response
please see attachment
The title of the work is: The design and characterization of a silver nanoparticle loaded, hydrogel forming wound dressing material. The design is not innovative, the characterization of the nanoparticles has not been performed, the activity on the wounds has not been studied. Only a modest part is reported on anticrobial activity.
In the attached response we have tried to spotlight the novelty of the work and we have explained why a wound model was not developed. We did include photographs, X ray data and size analysis data on nanoparticles but we understand the reviewer does not find these satisfactory. In the attached document we have tried to give a broader perspective of the novelty and importance of this work. Thanks
see attached.
